# Biosurfactants Induce Antimicrobial Peptide Production through the Activation of *Tm*Spatzles in *Tenebrio molitor*

**DOI:** 10.3390/ijms21176090

**Published:** 2020-08-24

**Authors:** Tariku Tesfaye Edosa, Yong Hun Jo, Maryam Keshavarz, In Seon Kim, Yeon Soo Han

**Affiliations:** 1Department of Applied Biology, Institute of Environmentally-Friendly Agriculture (IEFA), College of Agriculture and Life Sciences, Chonnam National University, Gwangju 61186, Korea; bunchk.2000@gmail.com (T.T.E.); yhun1228@jnu.ac.kr (Y.H.J.); mariakeshavarz1990@gmail.com (M.K.); 2Ethiopian Institute of Agricultural Research, Ambo Agricultural Research Center, Ambo 37, Ethiopia; 3Department of Agricultural Chemistry, College of Agriculture and Life Sciences, Chonnam National University, Gwangju 61186, Korea; mindzero@chonnam.ac.kr

**Keywords:** antimicrobial peptides, biosurfactants, glycolipids, immunity, lipopeptides, mealworm, *TmSpatzle*

## Abstract

Biosurfactant immunomodulatory activities in mammals, nematodes, and plants have been investigated. However, the immune activation property of biosurfactants in insects has not been reported. Therefore, here, we studied the defense response triggered by lipopeptides (fengycin and iturin A), glycolipids (rhamnolipid), and cyclic polypeptides (bacitracin) in the coleopteran insect, mealworm *Tenebrio molitor*. The in vitro antimicrobial activities against Gram-positive (*Staphylococcus aureus*) and Gram-negative (*Escherichia coli*) bacteria and fungi (*Candida albicans*) were assessed by mixing these pathogens with the hemolymph of biosurfactant-immune-activated larvae. *E. coli* growth was remarkably inhibited by this hemolymph. The antimicrobial peptide (AMP) induction results also revealed that all biosurfactants tested induced several AMPs, exclusively in hemocytes. The survivability analysis of *T. molitor* larvae challenged by *E. coli* (10^6^ CFU/µL) at 24 h post biosurfactant-immune activation showed that fengycin, iturin A, and rhamnopid significantly increased survivability against *E. coli*. Biosurfactant-induced *TmSpatzles* activation was also monitored, and the results showed that *TmSpz3* and *TmSpz-like* were upregulated in the hemocytes of iturin A-injected larvae, while *TmSpz4* and *TmSpz6* were upregulated in the fat bodies of the fengycin-, iturin A-, and rhamnolipid-injected larvae. Overall, these results suggest that lipopeptide and glycolipid biosurfactants induce the expression of AMPs in *T. molitor* via the activation of spätzle genes, thereby increasing the survivability of *T. molitor* against *E. coli*.

## 1. Introduction

Biosurfactants are surface-active biomolecules containing hydrophobic and hydrophilic moieties that are produced by bacteria, fungi, and yeast [1]. Unlike chemically synthesized surfactants, biosurfactants are generally categorized based on their microbial origin and chemical composition [2]. Chemically, biosurfactants are categorized into glycolipids (rhamnolipids) [3], trehalolipids [4], sophorolipids [5], lipopeptides and lipoproteins (surfactin, fengycine, and iturin), fatty acids, phospholipids and neutral lipids [5], polymeric, and particulate biosurfactants [6].

Owing to their safe properties, such as their low toxicity [7], high degree of biodegradability [8], high foaming capacity [9], and optimal activity under extreme environmental conditions [10], biosurfactants have received recent attention for their different applications in various fields, including medicine, food, bioremediation of toxic pollutants, and biopesticides.

The application of biosurfactants in agriculture has been repeatedly reported in plant disease control (as they exhibit antifungal and antibacterial activity [11,12]) and plant growth promotion [13]. More specifically, lipopeptide biosurfactants exhibit growth inhibition of phytopathogenic fungi, like *Fusarium* spp., *Aspergillus* spp., and *Biopolaris sorokiniana* [14]. Moreover, lipopeptides isolated from *Brevibacilis brevis* have demonstrated strong antibacterial and antifungal properties [15]. Interestingly, rhamnolipid showed the ability to stimulate plant immunity, reducing plant pathogen infection [16]. Based on their antimicrobial properties, it could be suggested that microbial biosurfactants can be exploited for phytopathogen control.

In medicine, lipopeptides have received the utmost attention owing to their high surface activity and antibiotic potential, which aid their applications as antibiotics, antitumor agents, and immunomodulators [17,18,19]. Lipopeptides have strong activity against multidrug-resistant bacteria, like *Escherichia coli* and *Staphylococcus aureus* [20]. Additionally, the studied antibacterial and biofilm disruption properties of biosurfactants (rhamnolipids and sophorolipids) for *E. coli* NCTC 10418 showed their ability to attach to surfaces, and they inhibit microbial growth and the resulting biofilm formation [21]. Iturin is a well-known lipopeptide biosurfactant reported to have antimicrobial activity [22]. Structurally, it is a cyclic peptide of seven amino acids linked to a fatty acid (*β*-amino) chain that can vary in length from C-14 to C-17 carbon molecules. These molecules increase iturin’s potency for plasma membrane disruption and increase the electrical conductance of biomolecular lipid membranes [23]. Therefore, iturin has been proposed as an antifungal agent in medicine. Additionally, fengycin, a lipopeptide containing a peptide chain of 10 amino acids linked to a fatty acid chain [24], has been reported as an antifungal agent [25].

The immunomodulatory activities of biosurfactants in mammals [17,26,27], *Caenorhabditis elegans* [28], and plants [29] have been investigated. More specifically, certain bacterial lipopeptides have been shown to be potent non-toxic, non-pyrogenic immunological adjuvants when mixed with conventional antigens [30]. For example, low-molecular-weight antigen iturin AL, herbicolin A, and microcystin coupled with poy-L-lysine enhance the humoral immune response in rabbits and chickens [17]. Additionally, emulsan has been approved for use as an adjuvant owing to its immunomodulatory potential in the human body.

In the mammalian innate and adaptive immune systems, conserved bacterial and viral components are detected by pattern recognition receptors. Specifically in humans, toll-like receptors have been identified as receptors for signals generated by lipopolysaccharides (LPS) and lipoproteins/lipopeptides (LPs/LPTs) to activate the immune response [31]. Conversely, the activation of insect immune-related genes begins when pathogen-associated molecular patterns (PAMPs) are recognized by germ-line-encoded receptors and soluble proteins [32]. PAMPs are conserved molecules on bacterial or fungal cell walls and include peptidoglycans (PGNs) on Gram-positive or Gram-negative bacteria and β-1,3-glucans on fungi [32]. Therefore, in insects, the lysine-containing PGN from Gram-positive bacteria and β-1,3-glucans from fungi are recognized by peptidoglycan recognition proteins (PGRPs) and Gram-negative binding protein (GNBP), respectively [33,34,35]. In general, the recognition of these proteins leads to the activation of serine protease cascades, including modular serine protease (MSP), Spätzle-processing enzyme-activating enzyme (SAE), and Spätzle-processing enzyme (SPE) [36,37], and eventually results in the cleavage of prospätzle. Finally, the processed spätzle binds to toll-like receptors, which play a key role in the insect immune response [36].

The recognition of these molecules by PGRPs or GNBP is selectively based on their structure. For instance, structurally, PGNs consist of carbohydrate chains connected to β (1-4), N-acetylglucosamine, and N-acetylmuramic acid sugars cross-linked by short peptide chains with alternating L- and D-amino acids [38]. Interestingly, D-amino acids are also recognized by receptors and enzymes in mammals to provide either a direct toxic response or indirect actions through modulating antimicrobial peptides [39]. Lipopeptides, the most commonly known biosurfactants, also consist of lipids connected to a peptide backbone containing a mix of L- and D-amino acids [40]. Additionally, rhamnolipids, one of the well-studied glycolipids, are composed of one or more monosaccharide moieties bound by a glycosidic linkage to a hydrophobic moiety [41]. Considering their peptide or sugar backbone compositions, lipopeptides and glycolipids are supposed to induce the innate immune response. Based on these reports of biosurfactant applications in plants and animals, it can be said that biosurfactants have both antimicrobial and immune-eliciting properties. However, the biosurfactant-triggered defense response in insects is yet to be studied.

Therefore, in this study, the induction of *TmSpatzle* by lipopeptides (fengycin and iturin A), polypeptide (bacitracin), and glycolipid (rhamnolipid) was studied. To the best of our knowledge, this is the first report indicating that lipopeptides (fengycin and iturin A) and glycolipids (rhaminolipds) increase the survivability of the mealworm *Tenebrio molitor* by activating different antimicrobial peptides through the activation of *TmSpz4*, *TmSpz6*, *TmSpz3,* or *TmSpz-like* in *T. molitor*.

## 2. Results

### 2.1. AMP Activity in the Hemolymph of Biosurfactant-Injected T. molitor Larvae

To determine the in vitro AMP activity against Gram-positive and Gram-negative bacteria and fungi, the hemolymph of biosurfactant-immune activated larvae was mixed with *E. coli*, *S. aureus*, and *C. albicans*. All the tested biosurfactants significantly (*p* < 0.05) reduced the growth of *E. coli* compared with PBS (Figure 1). Conversely, there was no difference between the control group (PBS) and biosurfactants group regarding growth of Gram-positive bacteria and fungi (Figure 1).

### 2.2. Biosurfactants Induced AMP Expression in T. molitor Hemocytes

To further understand how *E. coli* growth was inhibited by the hemolymph of biosurfactant-immune activated *T. molitor* larvae, the AMP expression level was assessed after injecting biosurfactants into *T. molitor* larvae. RNA was extracted 24 h post injection, and qRT-PCR was conducted using *T. molitor* AMP-specific primers. All tested biosurfactants induced different types of AMPs in hemocytes (Figure 2A,B,D). Particularly, iturin A upregulated the expressions of *TmTen-1* (4-fold), *TmTen-3* (14.67-fold), *TmTen-4* (15.72-fold), *TmAtt-1b* (14.49-fold), *TmCol-1* (6-fold), *TmDef-1* (12-fold), and *TmDef-2* (10-fold) (Figure 2C). Similarly, bacitracin (*TmTen-3*, *TmDef-1,* and *TmDef-2*), fengycin (*TmTen-2, TmTen-4,* and *TmAtt-1a*) (Figure 2B), and rhamnolipids (*TmTen-1*, *TmTen-2, TmTen-3*, *TmTen-4*, *TmDef-1*, and *TmDef-2*) slightly upregulated the expression of AMPs (Figure 2D). In hemocytes, *TmTen-3*, *TmDef-1*, and *TmDef-2* were upregulated by all tested biosurfactants, excluding fengycin. Likewise, *TmTen-2* was upregulated only by fengycin and rhamnolipids, while *TmAtt-1b* was upregulated by iturin A and fengycin. Conversely, in the fat bodies, most of the AMPs were either downregulated or remained constant like those of the control group. In the gut, except bacitracin and iturin A, which remarkably downregulated AMPs, the expressions of almost all the AMPs were similar to the control.

### 2.3. Biosurfactants Increased T. molitor Larvae Survivability against E. coli

To confirm if biosurfactant-activated AMPs could increase survivability against *E. coli*, a *T. molitor* larvae survivability assay was conducted by challenging the insect with *E. coli* (10^6^ CFU/µL) at 0, 12, and 24 h post immune activation by biosurfactants. Interestingly, fengycin, iturin A, and rhamnolipids significantly (*p* < 0.05) increased the survivability in 24 h post immune activated groups (Figure 3). Fengycin, iturin A, and rhamnolipids increased survivability against *E. coli* by 33.72, 34.48, and 28.75%, respectively. However, larvae challenged by *E. coli* at both 0 and 12 h post immune activation by biosurfactant did not show significant survivability when compared with the control group. This may suggest that AMP expression was regulated 24 h post biosurfactant-immune activation. Thus, survivability results confirm the AMP expression observed in this study.

### 2.4. Biosurfactants Activated the Expression of Spätzle Genes in T. molitor

To understand if biosurfactants induce AMP expression via the activation of *spätzle* genes in *T. molitor*, biosurfactant-induced activation of *T. molitor* spätzle was first examined and screened. The results revealed that *TmSpz3* and *TmSpz-like* were upregulated in hemocytes by iturin A, while *TmSpz4* and *TmSpz6* were upregulated in the fat bodies by fengycin, iturin A, and rhamnolipids (Figure 4). The other spatzles remained constant or downregulated, compared to the PBS-injected group. In the bacitracin-injected larvae, no *spätzle* was significantly upregulated. Comparatively, higher fold changes in *TmSpz4* and *TmSpz6* expressions were observed in rhamnolipid-injected groups.

### 2.5. Effect of TmSpz Gene Silencing on the Expression of AMPs in T. molitor

To study whether activated spatzles have a role in biosurfactant-induced AMP expression in *T. molitor*, the screened *spätzle* genes (*TmSpz3*, *TmSpz4*, *TmSpz6*, and *TmSpz-like*) were silenced. Then, we checked and confirmed the gene silencing levels of *TmSpz3* (Figure 5A), *TmSpz-like* (Figure 5C), *TmSpz4* (Figure 6A), and *TmSpz6* (Figure 7A). The biosurfactants were then injected into the gene-silenced larvae, respectively. The quantification of AMP expression showed that *TmSpz3*, *TmSpz4*, *TmSpz6*, and *TmSpz-like* positively regulated the expression of most of the AMPs in the whole body. Specifically, the expressions of *TmTen-2, -4*, *TmAtt-1a*, *TmAtt-1b*, *TmAtt-2*, *TmCol-1,* and *TmCol-2* were significantly downregulated in the *TmSpz3*-silenced and iturin A-injected larvae (Figure 5B). The expressions of *TmTen-2, TmCol-1,* and *-2* were positively regulated by *TmSpz-like* (Figure 5D).

Likewise, *TmSpz4* knock-down remarkably reduced the expression of AMPs induced by fengycin (*TmTen-2, TmTen-3*, *TmTen-4*, *TmAtt-1a, TmCol-2*, and *TmDef-2*) (Figure 6B) and iturin A (*TmTen-2, TmTen-3, TmTen-4*, *TmDef-2, TmAtt-1a, TmAtt-1b*, *TmAtt-2*, and *TmCol-2)* (Figure 6C) in the whole body. The expressions of most of the AMPs induced by rhamnolipid were negatively regulated by *TmSpz4* in (Figure 6D).

The deletion of *TmSpz6* resulted in a significant (*p* < 0.05) reduction in the AMP expression induced by fengycin (*TmTen-1, TmTen-4*, *TmDef-2, TmAtt-1a, TmAtt-1b, TmCol-1*, and *TmCol-2*) (Figure 7B), iturin A (*TmTen-2, TmTen-4*, *TmAtt-1b, TmCol-1*, and *TmCol-2*) (Figure 7C), and rhamnolipid (*TmTen-1, TmTen-4*, *TmDef-1, TmDef-2, TmAtt-1a, TmAtt-1b, TmCol-1*, *TmCol-2*, and *TLP-2*) (Figure 7D).

### 2.6. T. molitor Larvae Survivability Study

To further understand if the biosurfactants increased the survivability of *T. molitor* via *spätzle* gene activation and AMP induction against microbial challenge, *TmSpzs* were individually silenced (Figure 8A,C, Figure 9A and Figure 10A), and each biosurfactant was injected into the gene-silenced larvae. Next, 24 h post biosurfactant injection, *E. coli* was injected. The results revealed that in the *TmSpz3* and -*like* gene-silenced larvae, iturin A injection did not increase the survivability of larvae against *E. coli* infection, whereas in the ds*EGFP* injected larval group (control), iturin A significantly increased the survivability (Figure 8B,D). Meanwhile, fengycin and iturin A increased the survivability of larvae injected with ds*EGFP*, while *TmSpz4* gene-silenced larvae were susceptible to *E. coli* (Figure 9B,C). Conversely, rhamnolipid showed no significant induction of larval survivability among the *TmSpz4* gene-silenced and ds*EGFP*-injected larvae (Figure 9D). Similarly, although fengycin and rhamnolipid significantly (*p* < 0.05) increased control group survivability, the ds*TmSpz6*-injected larvae were susceptible to *E. coli* (Figure 10B–D).

## 3. Discussion

Biosurfactants have been repeatedly shown to exhibit antimicrobial activity [42,43,44]. In addition, the immune response modulators of biosurfactants have been investigated in mammalian and nematode *C. elegans* [26,27,28]. Therefore, based on the immunomodulatory reports, it could be hypothesized that biosurfactants could activate the immune response of *T. molitor* against microorganisms. Thus, to test this postulation, the in vitro growth inhibitions of Gram-positive and Gram-negative bacteria and fungi were performed by mixing microbial cells with the hemolymph of biosurfactant-immune activated larvae. Excitingly, the results revealed that all the tested biosurfactants significantly reduced the CFU of the Gram-negative bacteria *E. coli*, with no difference observed between the control and biosurfactants against Gram-positive bacteria and fungi. These results suggest that the tested biosurfactants specifically inhibit the growth of Gram-negative bacteria via activating the production of AMPs in *T. molitor* hemolymph.

In insects, AMPs are constitutively produced in fat bodies and hemocytes and are then released into the hemolymph in response to infection or immune elicitors after the recognition PAMPs [45,46,47]. In *T. molitor*, the identified AMPs include tenecins (-1, -2, -3, -4) [48,49,50], attacins [51], defensins [52], and coleopterins [53]. In general, AMPs are known to act against Gram-positive and -negative bacteria and fungi [48,54,55] by membrane disruption, interference with bacterial metabolism, and targeting of cytoplasmic components [47,56,57]. Therefore, based on our in vitro results, it could be suggested that biosurfactants induce the production of active AMPs against *E. coli*. Interestingly, besides their antimicrobial properties, biosurfactants have the ability to elicit mealworm immune responses against microorganisms.

The inhibition of *E. coli* CFU by the hemolymph of biosurfactant-immune triggered *T. molitor* larvae was a call for concern. As explained earlier, AMPs form the main immune response produced during infection or immune elicitation. Therefore, AMP expression levels were determined by injecting biosurfactants into *T. molitor* larvae. Various AMPs were induced by the injection of biosurfactants, suggesting that biosurfactants could regulate the innate immunity of *T. molitor* through the induction of AMPs in hemocytes. However, using a different mechanism, *B. licheniformis*-derived [58] and rhamnolipid biosurfactants [59] have also demonstrated the ability for modulation of immune responses in *Labeo rohita*. Moreover, rhamnolipid displayed immunostimulatory properties against human monocyte cells by improving the secretion of the cytokine tumor necrosis factor [60].

The major concern here is how the host recognizes these lipopetides or glycolipids to activate the immune response. Even though the known pattern recognition receptors of D-amino acids have not been identified, D-amino acids are asymmetrically recognized by some other receptors and enzymes in mammals to provide either direct toxic responses or indirect actions through the modulation of antimicrobial peptides [39]. Additionally, Li Hao and his colleagues described that the presence of D-amino acids in the peptide structure can play a key role in mammalian receptor recognition [61]. Based on the composition of the different L- and D-amino acids, including D-amino acids (Tyr, Orn, and Thr, Ala) in fengycin and the D-amino acids (Tyr, Asn, and Asn) of the cyclic lipopeptide iturin A [62], the recognition of these biosurfactants could be attributable to certain recognition receptors that activate AMPs.

Toll receptors have been identified and known to recognize evolutionarily conserved bacterial components such as LPS, LP, LPT, and PGN in mammals [31,63]. Moreover, it has been reported that CD14 facilitates LP recognition by toll-like receptors 2 and 1 in humans [64,65,66]. Similarly, LPS binding protein (LBP) is a serum factor known to have the ability to bind pathogens and initiate the innate immune response [67]. Moreover, it has been reported that LP could enhance TLR2-mediated activation, a promising strategy for vaccine development [68].

Contrarily, in insects, during both embryogenesis and immunity, inactive spätzle protein is synthesized and secreted from the cell, containing a pro domain and a C-terminal region [69,70]. During infection, microbe-associated molecular patterns, such as LPS and lipoteichoic acid (LTA), are recognized by PGN recognition protein and GNBP1 [33], leading to the initiation of the proteolytic cascade, serine protease, and SPE [36]. Therefore, this process finally activates the initially inactive spätzle. Extrapolating to the mammalian and insect immune activation systems, the *bacillus* LP (fengycin and iturin A) and glycolipid (rhamnolipid) are found activating the *TmSpatzle* genes in *T. molitor.* It should be noticed that the recognition receptor for these specific biosurfactants is not well known.

The effect of biosurfactants in the survivability of the *T. molitor* Spätzle gene-silenced group was examined, and the *TmSpz* gene-silenced larvae showed susceptibility to *E. coli*, while the control group larvae significantly survived. This indicates that the biosurfactant-induced immune response occurs via the activation of *TmSpz* genes. Most of the AMPs suppressed in the absence of spätzle genes including *TmSpz4* (*TmTen-2*, *TmAtt-1a*, and *TmCol-2*), *TmSpz3* (*TmTen-2*, *TmTen-4*, *TmAtt-1b*, *TmCol-1*, and *TmCol-2*), and *TmSpz6* (*TmTen-2, TmTen-4, TmDef-2, TmAtt-1a, TmAtt-1b, TmCol-1*, and *TmCol-2*) are the well-known AMPs active against Gram-negative bacteria [48,71,72]. Most glycine-rich AMPs are highly specific and effective against groups of Gram-negative bacteria [73]. Supporting our current survivability and AMP expression study results, attacins in *Hyalophora cecropia* [74], *TmTen-4* in *T. molitor* [48], and *TmTen-2* in *T. molitor* [71], which are glycine-rich AMPs, have been reported as being particularly effective against Gram-negative bacteria [71]. Similarly, in our recent publication we have reported *TmSpz6* is important in regulating the AMPs production in response to *E. coli* [75]. Therefore, it can be said that the biosurfactant-activated AMPs enable larvae to be protected against *E. coli*. Altogether, our results revealed that LP and glycolipid biosurfactants induce AMP expression in *T. molitor* through spätzle gene activation, thereby increasing *T. molitor* larvae survivability against *E. coli*.

## 4. Materials and Methods

### 4.1. Insect Culture

The coleopteran insect mealworm, *Tenebrio molitor*, was maintained at 27 ± 1 °C, 60% ± 5% relative humidity, under dark conditions, with an artificial diet (170 g of whole-wheat flour, 20 g of fried bean powder, 10 g of soy protein, 100 g of wheat bran, 200 mL of sterile water, 0.5 g of chloramphenicol, 0.5 g of sorbic acid, and 0.5 mL of propionic acid). Larvae at the 10th to 12th instar were used for experiments [76].

### 4.2. Source of Biosurfactants

The biosurfactants (bacitracin, fengycin, iturin A, and rhamnolipids) used in this study were purchased from Sigma Aldrich (St. Louis, MO, USA) (Table 1). All the biosurfactants were in powder form.

### 4.3. Preparation of Microorganisms

Gram-negative bacteria (*E. coli* K12), Gram-positive bacteria (*S. aureus* RN4220), and fungi (*Candida albicans*) were used in this study. These microorganisms were cultured in Luria-Bertani (LB; *E. coli and S. aureus*) and Sabouraud dextrose (*C. albicans*) broths at 37 °C overnight, subcultured at 37 °C for 3 h. Then, the microorganisms were harvested and washed two times by centrifugation at 3500 rpm for 10 min in phosphate-buffered saline (PBS; pH 7.0). They were then suspended in PBS, and their concentrations were measured at OD600. Finally, 10^6^ cells/μL of *E. coli*, *S. aureus*, and 5 × 10^4^ cells/μL of *C. albicans* were injected separately [77].

### 4.4. AMPs Activity Study

Each biosurfactant was dissolved by vortexing in the 1x PBS at 1μg/μL (biosurfactant/PBS). Then, the stock solution of each biosurfactant was diluted to the targeted concentration (100, 200, 300, and 1000 ng/µL). Late instars larvae of *T. molitor* were injected with 300 ng/µL of each biosurfactant. During each injection, the solution was well vortexed to insure uniform distribution of the particle. This concentration was selected based on the previous preliminary experiment. Briefly, we conducted the preliminary experiment with 100, 200, 300, and 1000 ng/µL biosurfactants injection, and the highest AMPs activity was achieved at 300 ng/µL. 1X PBS (0.1 mM sodium phosphate, 2.5 mM potassium chloride, pH 7.2) was injected as control. The injections were done using disposable needles mounted onto a micro applicator (Picospiritzer III micro dispense system, Parker Hannifin, Hollis, NH). The injected larvae were maintained on artificial diet under standard rearing conditions. Twenty-four hours post injection, the hemolymph was extracted in to 1X PBS on ice. Then, the hemolymph was centrifuged at 15,000 rpm and 4 °C for 10 min. The supernatant was retrieved and boiled at 100 °C for 10 min, then centrifuged at 15,000 rpm and 4 °C for 10 min, and then the supernatant transferred to a new tube. Finally, peptide concentration was determined by Ephoch at OD220. The hemolymph sample was diluted by 1X PBS and 100 µg mixed with *E. coli* (10^6^ CFU). The total 200 µL mixture (10 µL hemolymph sample, 10 µL *E. coli,* and 180 µL PBS) was incubated at 37 °C for 1 h under shaking conditions (200 rpm). Then, 10^−5^ serial dilution was done, and 100 µL of the mixture was plated on LB agar media and incubated at 37 °C for 16 h. The assay was done in triplicate for each biosurfactant, and the colony per plate for each biosurfactant was counted.

### 4.5. RNA Extraction, cDNA Synthesis, and Expression of AMPs

Each biosurfactant was injected at 300 ng/μL in to late larvae of *T. molitor.* 1X PBS was injected into the control group larvae. Twenty-four hours after biosurfactant injection, the whole-body samples from 20 larvae were injected with each biosurfactant to 500 μL of guanidine thiocyanate RNA lysis buffer, which was prepared from 0.5 M EDTA, 1 M MES buffer, 3 M guanidine thiocyanate, 200 mM sodium chloride, phenol red (40 μL), 25 μL of Tween-80, 250 μL of acetic acid glacial (for pH 5.5), and 500 μL of isoamyl alcohol. Samples were homogenized using a homogenizer machine (Bertin Technologies, France) for 20 s. RNA extraction was done following the protocol [76].

Expression patterns of fourteen AMPs—TmTencin-1 (TmTen-1), TmTencin-2 (TmTen-2), TmTencin-3 (TmTen-3), TmTencin-4 (TmTen-4), TmAttacin-1a (TmAtt1a), TmAttacin-1b (TmAtt-1b), TmAttacin-2 (TmAtt-2), TmDefensin-1 (TmDef-1), TmDefensin-2 (TmDef-2), TmColptericin-1 (TmCol-1), TmColptericin-2 (TmCol-2), TmCecropin-2 (TmCec-2), TmThaumatin like protein-1 (TmTLP-1), and TmThaumatin like protein-2 (TmTLP-2)—were investigated by quantitative real-time polymerase chain reaction (qRT-PCR) using specific primers (Table 2), with AccuPower^®^ Greenstar^TM^ qPCR PreMix (Bioneer, Daejeon, Korea) and the AriaMx Real-Time PCR System (Agilent Technologies, USA). The qRT-PCR conditions were as follows: pre-denaturation at 95 °C for 5 min, followed by 40 cycles of denaturation at 95 °C for 15 s, and annealing and extension at 60 °C for 30 s. Primers designed from *T. molitor* ribosomal protein L27a (TmL27a) were used as internal control.

### 4.6. Biosurfactant-Triggered Immune Survivability of T. molitor

Thirty young instars (9–12 instars) larvae of *T. molitor* were injected with 300 ng/µL of each biosurfactant, and 1X PBS as control. At different time points (0, 12, and 24 h) post injection, *E. coli* (10^6^ CFU) were injected in to the immune-triggered and PBS-injected larvae. In the 0 h group, biosurfactants and *E. coli* were injected at the same time. The experiment was performed in triplicate (10 larvae/replication). Mortality data were counted daily for 10 d.

### 4.7. Screening of Biosurfactant-Activated T. molitor Spatzle Genes

Biosurfactants were injected in to *T. molitor* larvae, and 24 h post injection, three tissues—the hemocytes, fat bodies and guts—were dissected, RNA extracted, and cDNA synthesized following the abovementioned protocol. Using gene-specific primers, the expressions of nine *T. molitor* Spätzle genes following biosurfactant injection were analyzed. The expression was quantified by qRT-PCR.

### 4.8. Effect of Tm Spätzle Gene Silencing on the Expression of AMPs

To synthesize the double-stranded RNA of the *TmSpz* genes, forward and reverse primers containing the T7 promoter sequence at their 5′ ends were designed using the SnapDragon-Long dsRNA design software (Harvard Medical School, Boston, MA, USA, https://www.flyrnai.org/cgibin/RNAi_find_primers.pl). The PCR product was amplified using AccuPower^®^ Pfu PCR PreMix with TmSpz_Fw and TmSpz_Rv under the following cycling conditions: an initial denaturation step at 94 °C for 2 min, followed by 35 cycles of denaturation at 94 °C for 30 s, annealing at 53 °C for 30 s, and extension at 72 °C for 30 s, with a final extension step at 72 °C for 5 min. The PCR products were purified using the AccuPrep PCR Purification Kit (Bioneer, Daejeon, Korea), and dsRNA was synthesized using the Ampliscribe^TM^ T7-Flash^TM^ Transcription Kit (Epicentre Biotechnologies, Madison, WI, USA), according to the manufacturer’s instructions. After synthesis, the dsRNA was purified by precipitation with 5 M ammonium acetate and 80% ethanol. Subsequently, it was quantified using an Epoch spectrophotometer (BioTek Instruments, Inc., Winooski, VT, USA). As a control, the dsRNA of enhanced green fluorescent protein (ds*EGFP*) was synthesized, and all samples were stored at −20 °C until use.

To study whether the biosurfactants induced AMPs expression in *T. molitor* via the activation of Spätzle genes, *TmSpz* genes—those expressed by biosurfactants (*T. molitor spätzle 3 (TmSpz3), spätzle 4* (*TmSpz4*), *spätzle 6* (*TmSpz6*), and *spätzle like* (*TmSpz-like*))—were silenced by injecting the synthesized dsRNA (1 µg/larvae) of each gene into the larvae hemocoel. ds*EGFP* was injected to a separate set of larvae that acted as negative control. The gene expression knockdown (>85%) was achieved at 3 d (*TmSpz3, TmSpz6*, and *TmSpz-like*) and 5 d (*TmSpz4*) post injection of each dsRNA. The respective biosurfactants per specific day were then injected in the gene-silenced larvae. 1XPBS was injected as control group. Twenty-four hours post biosurfactant injection, RNA was extracted from the whole body. These time points were selected based on the preliminary experiment of AMP expression. Subsequently, cDNA synthesis was conducted following the previously described protocol. qRT-PCR was then conducted using specific primers to analyze the temporal expression patterns of the fourteen AMPs.

### 4.9. Effect of TmSpz Genes Silencing on T. molitor Survivability

The survivability study was investigated after the *TmSpz3, TmSpz4, TmSpz6*, and *TmSpz-like* genes were silenced. After gene silencing, the following biosurfactants were injected: iturin A in *TmSpz3* and *TmSpz-like* gene-silenced larvae; fengycin, iturin, and bacitracin in *TmSpz4* and *TmSpz6* gene-silenced larvae. This grouping was based on the preliminary screening of *TmSpz* genes against biosurfactants. Twenty-four hours after biosurfactant injection, *E. coli* (10^6^ CFU) were injected in to the immune triggered larvae. The experiment was done in triplicate. Mortality data were counted daily for 10 days.

### 4.10. Data Analyses

The survivability data were subjected to analysis of variance (ANOVA) using SAS 9.4. Differences from the control were assessed using Tukey’s multiple range test (*p* < 0.05). Comparative AMP gene expression was calculated using delta delta *C*t (ΔΔ*C*t). The fold change from the internal control (*TmL27a*) and external control (PBS) was calculated using the formula 2^–(ΔΔ*C*t)^.

## 5. Conclusions

The antimicrobial properties of different biosurfactants from bacteria and fungi have been well studied. However, our current study revealed that fengycin and iturin A produced by *Bacillus subtilis* and rhamnolipids produced by *Pseudomonas aeruginosa* modulate the immunity of the yellow mealworm *T. molitor*. Generally, this study suggests that biosurfactants such as lipopeptides and glycolipids induce the expression of AMPs through activation of spätzle genes in hemocytes, thereby increasing the survivability of *T. molitor* larvae against *E. coli* (Figure 11). Based on our current results, these biosurfactants could be used as safe immune elicitors in the mass rearing of *T. molitor*.

## Figures and Tables

**Figure 1 ijms-21-06090-f001:**
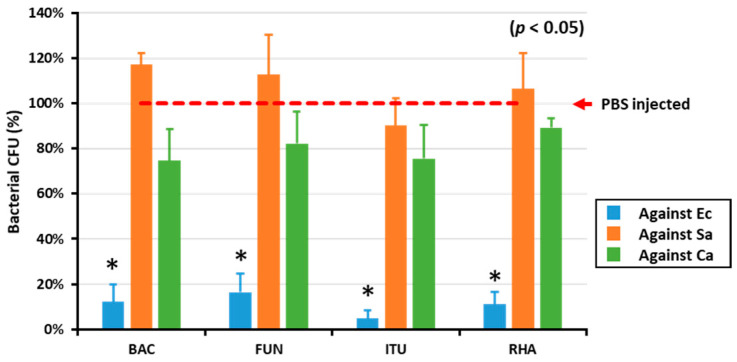
The in vitro AMP activity of biosurfactant-immune activated larvae against microbes. The in vitro AMP activities of biosurfactant-immune activated larvae against Gram-positive (*S. aureus*, Sa), Gram-negative bacteria (*E. coli*, Ec), and fungi (*C. albicans*, Ca) were measured by mixing the hemolymph of immune-activated larvae and the microbes. Excitingly, the results revealed that all the tested biosurfactants including bacitracin (BAC), fengycin (FUN), iturin A (ITU), and rhamnolipid (RHA) significantly (*p* < 0.05) reduced *E. coli* growth, compared with the PBS-injected control group (red dotted line).* significant difference (*p* < 0.05) of CFU, compared to the PBS-injected group.

**Figure 2 ijms-21-06090-f002:**
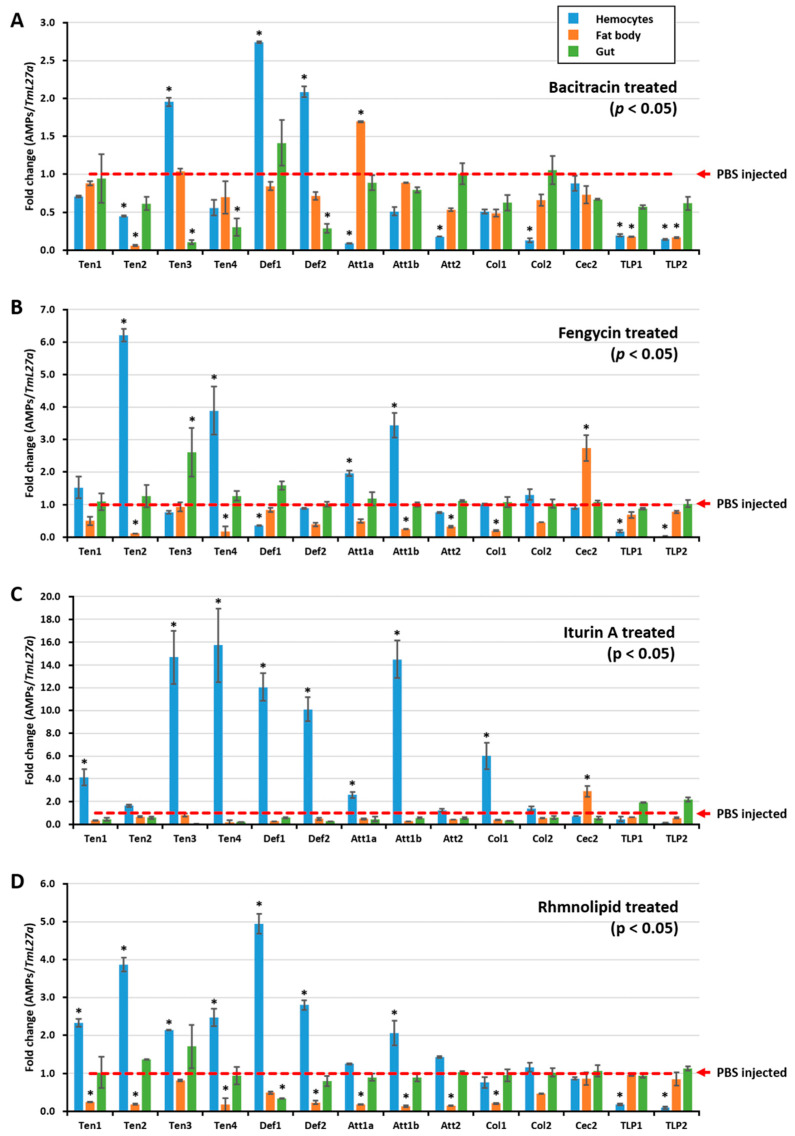
Biosurfactant-induced expression of antimicrobial peptides in *T. molitor*. The AMP expression levels were assessed 24 h after injecting either bacitracin (**A**), fengycin (**B**), itiurin A (**C**), or rhamnolipid (**D**) into *T. molitor* larvae. Three tissues—hemocytes, fat body, and gut—were then dissected. The expression of AMPs in each tissue was quantified by qRT-PCR, using specific primers. The expression results are presented as mean of three replicates. * significant difference (*p* < 0.05) in AMP expression, compared to the PBS-injected group. The AMP expression results revealed that all tested biosurfactants induced different types of AMPs in hemocytes.

**Figure 3 ijms-21-06090-f003:**
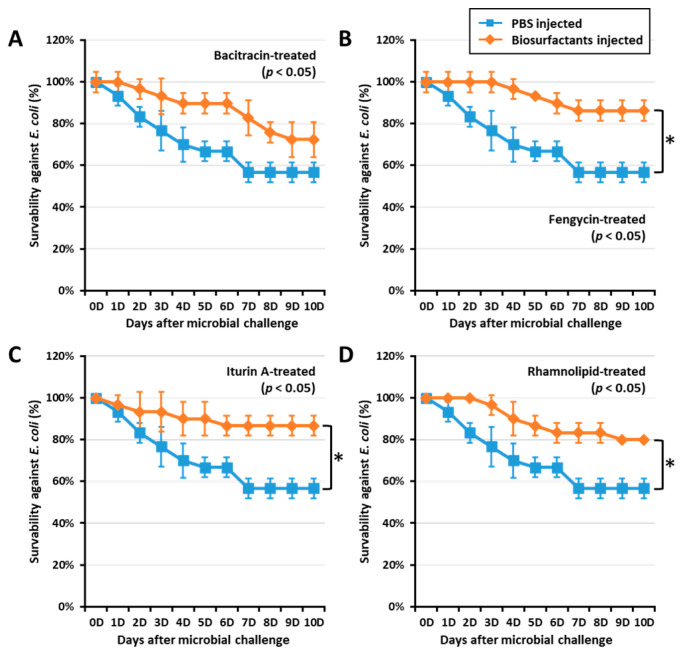
Effect of biosurfactants on *T. molitor* larvae survivability against *E. coli*. *T. molitor* larvae survivability against *E. coli* was assessed by challenging with *E. coli* (10^6^ CFU/μL) 24 h post immune activation by biosurfactants such as bacitracin (**A**), fengycin (**B**), iturin A (**C**), and rhamnolipid (**D**). Interestingly, fengycin, iturin A, and rhamnolipid significantly increased the survivability against *E. coli* by 33.72, 34.48, and 28.75%, respectively. The graph indicated by asterisks (*) shows significant difference at *p* < 0.05 following the Student–Newman–Keuls test.

**Figure 4 ijms-21-06090-f004:**
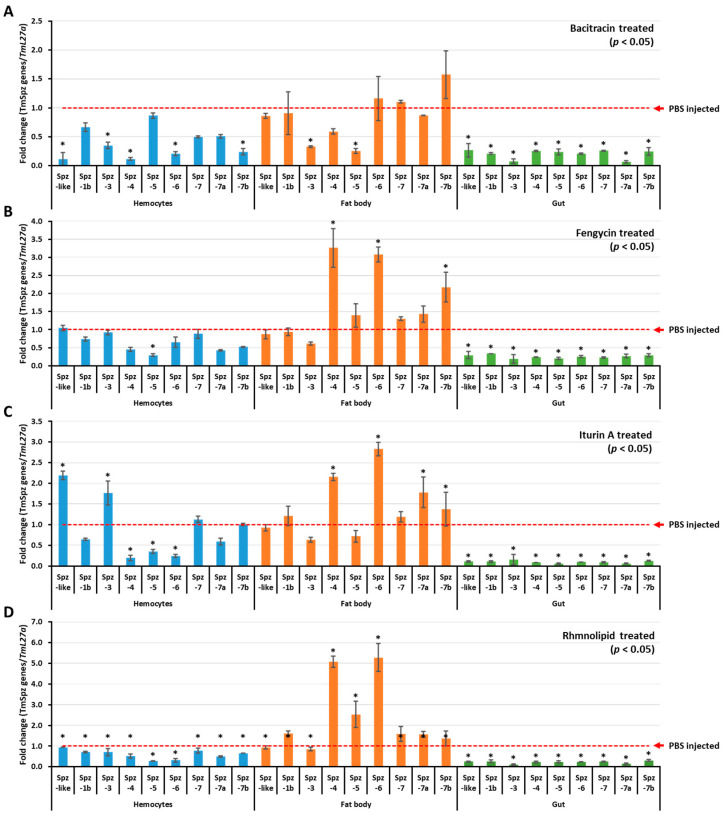
*Tm**Spzs* induction patterns in different tissues. The expressions of Spatzles gene in different tissues were examined by injecting either bacitracin (**A**), fengycin (**B**), iturin A (**C**), or rhamnolipid (**D**). Twenty-four hours post biosurfactant injection, three tissues—hemocytes, fat bodies, and gut—were dissected. The expression of spätzle in each tissue was quantified by qRT-PCR using specific primers. The data are the average of three replicates. * significant difference (*p* < 0.05) of Spatzles expression, compared to the PBS-injected group.

**Figure 5 ijms-21-06090-f005:**
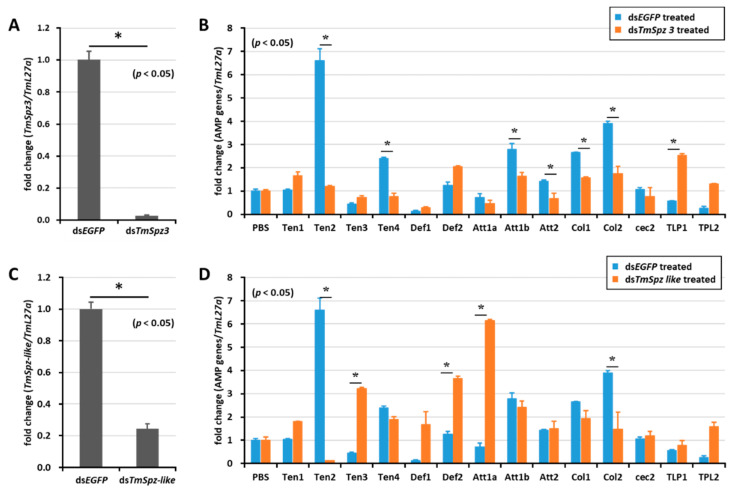
Induction of AMPs after gene silencing of *TmSpz-3* and *-like* and injection of iturin A. The expressions of *TmSpz3* (**A**) and *TmSpzlike* (**C**) were silenced, and iturin A was injected into the gene-silenced larvae. AMP expression was then examined in *T. molitor*, in the ds*TmSpz3* (**B**), and ds*TmSpz-like* (**D**)-injected larvae. The expression of AMPs was quantified by qRT-PCR, using specific primers. * indicates significant difference (*p* < 0.05) of AMPs expression, compared to the ds*EGFP*-injected group.

**Figure 6 ijms-21-06090-f006:**
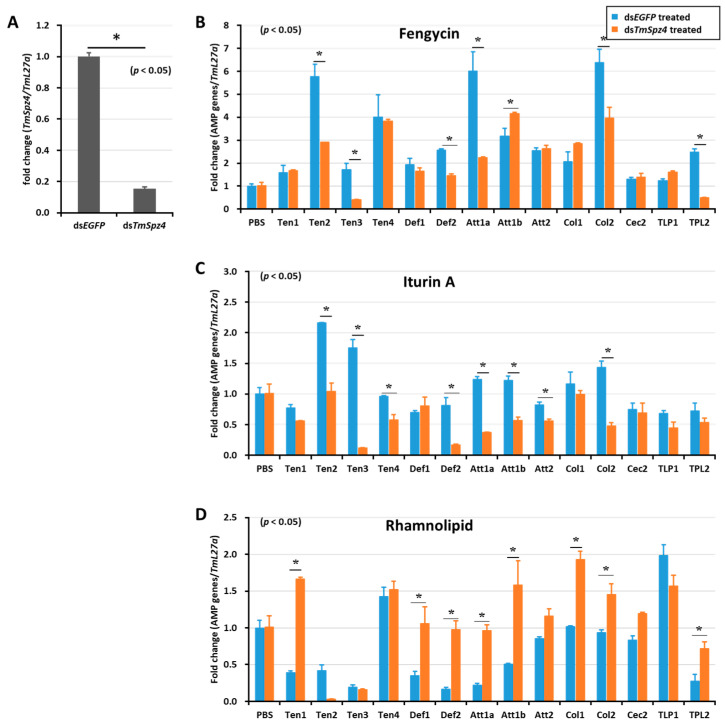
Effect of *TmSpz 4* gene silencing on AMP induction in *T. molitor* larvae. The expression of *TmSpz 4* was silenced (**A**), then either fengycin (**B**), iturin A (**C**), or rhamnolipid (**D**) was injected into the gene-silenced larvae. Whole-body AMP expression was examined after 9 h post biosurfactant injection. The expression of AMPs in each tissue was quantified by qRT-PCR, using specific primers. The data are the mean of three individual replicates. * significant difference (*p* < 0.05) in AMP expression, compared to the ds*EGFP*-injected group.

**Figure 7 ijms-21-06090-f007:**
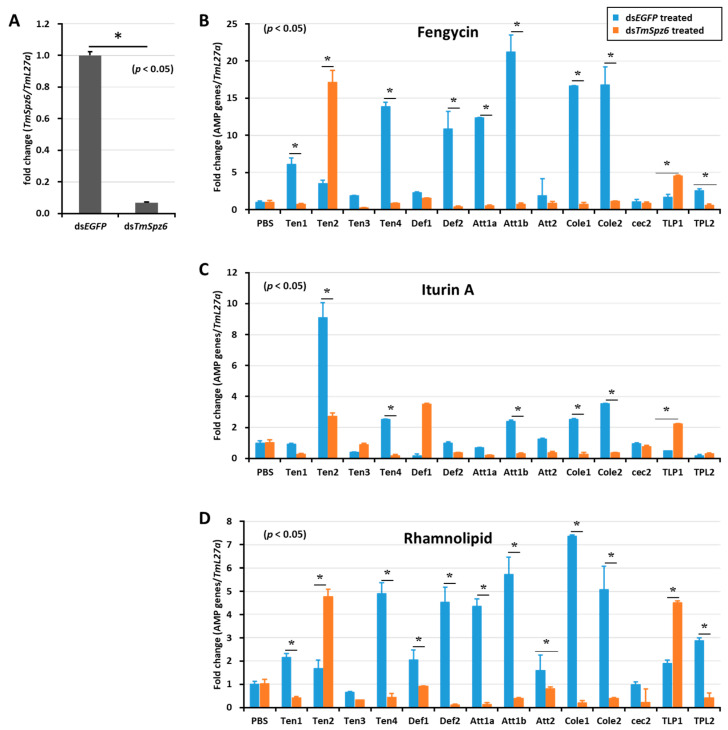
Effect of *TmSpz 6* gene silencing on AMP induction in *T. molitor* larvae. *TmSpz 6* expression was silenced (**A**), then either fengycin (**B**), iturin A (**C**), or rhamnolipid (**D**) was injected into the gene-silenced larvae. Whole-body AMP expression was examined after 9 h post biosurfactant injection. The expression of AMPs in each tissue was quantified by qRT-PCR, using specific primers. The data are the mean of three individual replicates. * significant difference (*p* < 0.05) in AMP expression, compared to the ds*EGFP*-injected group.

**Figure 8 ijms-21-06090-f008:**
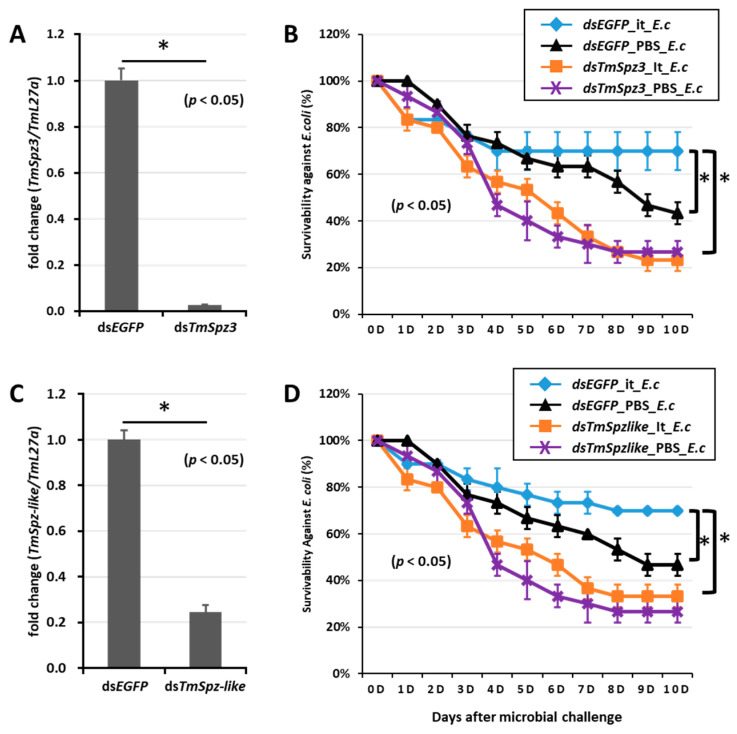
Survivability of *T. molitor* larvae against *E. coli* after *TmSpz-3* or -*like* gene silencing and iturin A injection. The *TmSpz-3* or -*like* gene were silenced, and the silencing efficiency was measured by qRT-PCR at 3 d post-injection (**A**,**C**). Iturin A (It) was then injected into the *TmSpz-3* (**B**) and -*like* (**D**) gene-silenced larvae. After 24 h of iturin A injection, *E. coli* (E.c) was injected into the gene-silenced larvae. ds*EGFP* and 1XPBS were injected as control for gene and iturin A- or microbe-injected groups, respectively. The data represent the mean of three biological replicate experiments. * significant differences between ds*TmSpz-3,* -*like,* and ds*EGFP*-treated and PBS-injected larval groups (*p* < 0.05).

**Figure 9 ijms-21-06090-f009:**
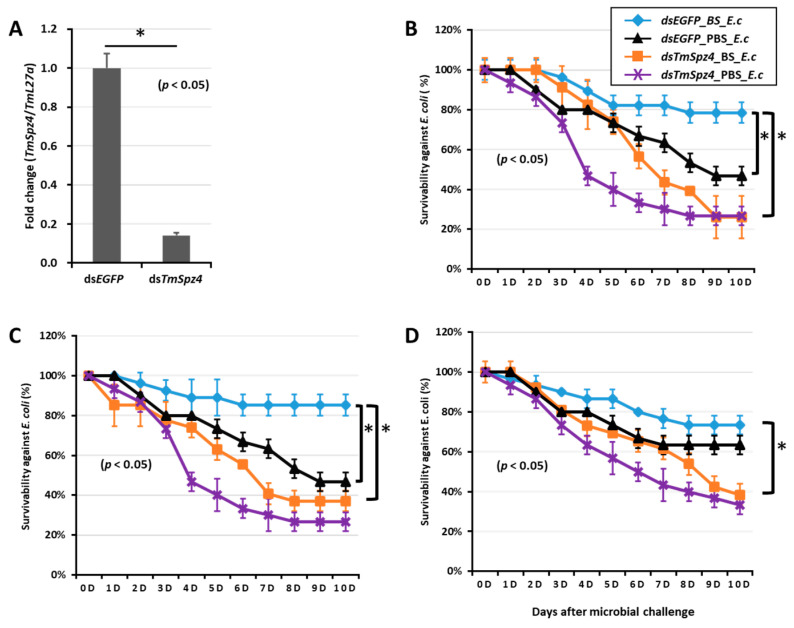
Survivability of *T. molitor* larvae against *E. coli* (E.c) after *TmSpz4* gene silencing and biosurfactant injection. The *TmSpz4* gene was silenced, and the silencing efficiency of *TmSpz4* mRNA was measured by qRT-PCR at 5 d post-injection (**A**). Then, biosurfactants (BS) including fengycin (**B**), iturin A (**C**), and rhamnolipid (**D**) were separately injected into the gene-silenced larvae. *E. coli* was then injected into the gene-silenced and biosurfactant-injected larvae. ds*EGFP* and 1XPBS were injected as control for the gene- and biosurfactant- or microbe-injected groups, respectively. The data represent the mean of three biological replicate experiments. * significant differences between ds*TmSpz4*- and ds*EGFP*-treated groups of biosurfactant- and PBS-injected larvae (*p* < 0.05).

**Figure 10 ijms-21-06090-f010:**
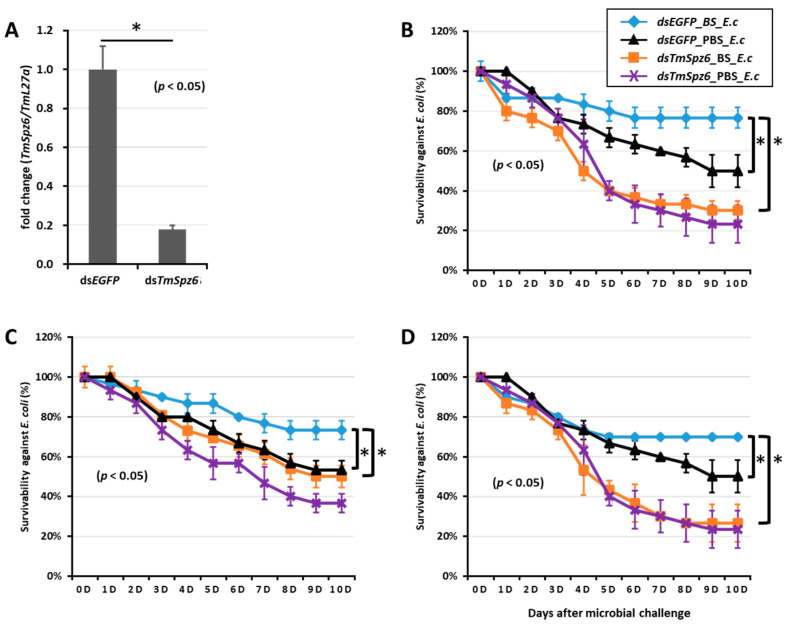
Survivability of *T. molitor* larvae against *E. coli* (E.c) after *TmSpz6* gene silencing and biosurfactant injection. The *TmSpz6* gene was silenced, and the silencing efficiency of *TmSpz6* mRNA was measured by qRT-PCR at 3 d post-injection (**A**). Then, biosurfactants (BS) including fengycin (**B**), iturin A (**C**), and rhamnolipid (**D**) were separately injected into gene-silenced larvae. *E. coli* was then injected to the gene-silenced- and biosurfactant-injected larvae. ds*EGFP* and 1XPBS were injected as control for the gene- and biosurfactant- or microbe-injected groups, respectively. The data represent the mean of three biological replicate experiments. * significant differences between ds*TmSpz6*- and ds*EGFP*-treated groups, biosurfactant-, and PBS-injected larvae (*p* < 0.05).

**Figure 11 ijms-21-06090-f011:**
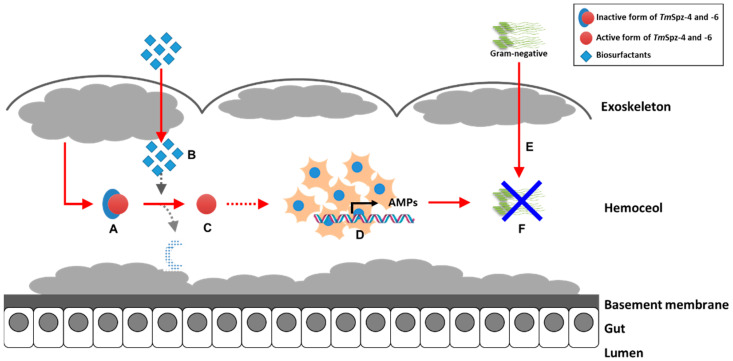
Proposed graphic summary of biosurfactants activity in the expression of AMPs through activation of *TmSpz-4* and *-6*. The pro-form of *Tm*Spzs (**A**) generated from fat body is activated by the injection of biosurfactants (**B**) to matured form of *Tm*Spzs (**C**). The activated *Tm*Spzs initiate the activation of intracellular toll signaling cascade in hemocytes to produce several AMPs (**D**), and when *E. coli* is injected into the *T. molitor* hemoceol (**E**), the produced AMPs protect the host from infection (**F**).

**Table 1 ijms-21-06090-t001:** List of biosurfactants, their types, and their biological sources.

Biosurfactants.	Type	Biological Source	Type	Purity
Bacitracin	cyclic lipopeptide	*Bacillus licheniformis*	Gram-positive	90%
Fengycin	lipopeptide complex	*Bacillus subtilis*	Gram-positive	>90%
Iturin A	cyclic lipopeptide	*Bacillus subtilis*	Gram-positive	>95%
Rhamnolipid	Glycolipids	*Pseudomonas aeruginosa*	Gram-negative	>95%

**Table 2 ijms-21-06090-t002:** Primers used in this study.

Name	Primer Sequences
TmTencin-1_Fw	5′-CAGCTGAAGAAATCGAACAAGG-3′
TmTencin-1_Rv	5′-CAGACCCTCTTTCCGTTACAGT-3′
TmTencin-2_Fw	5′-CAGCAAAACGGAGGATGGTC-3′
TmTencin-2_Rv	5′-CGTTGAAATCGTGATCTTGTCC-3′
TmTencin-3_Fw	5′-GATTTGCTTGATTCTGGTGGTC-3′
TmTencin-3_Rv	5′-CTGATGGCCTCCTAAATGTCC-3′
TmTencin-4_Fw	5′-GGACATTGAAGATCCAGGAAAG-3′
TmTencin-4_Rv	5′-CGGTGTTCCTTATGTAGAGCTG-3′
TmDefensin-1_Fw	5′-AAATCGAACAAGGCCAACAC-3′
TmDefencin-1_Rv	5′-GCAAATGCAGACCCTCTTTC-3′
TmDefencin-2_Fw	5′-GGGATGCCTCATGAAGATGTAG-3′
TmDefencin-2_Rv	5′-CCAATGCAAACACATTCGTC-3′
TmColoptericin-1_Fw	5′-GGACAGAATGGTGGATGGTC-3′
TmColoptericin-1_Rv	5′-CTCCAACATTCCAGGTAGGC-3
TmColoptericin-2_Fw	5′-GGACGGTTCTGATCTTCTTGAT-3′
TmColoptericin-2_Rv	5′-CAGCTGTTTGTTTGTTCTCGTC-3′
TmAttacin-1a_Fw	5′-GAAACGAAATGGAAGGTGGA-3′
TmAttacin-1a_Rv	5′-TGCTTCGGCAGACAATACAG-3′
TmAttacin-1b_Fw	5′-GAGCTGTGAATGCAGGACAA-3′
TmAttacin-1b_Rv	5′-CCCTCTGATGAAACCTCCAA-3′
TmAttacin-2_Fw	5′-AACTGGGATATTCGCACGTC-3′
TmAttacin-2_Rv	5′-CCCTCCGAAATGTCTGTTGT-3
TmCecropin-2_Fw	5′-TACTAGCAGCGCCAAAACCT-3′
TmCecropin-2_Rv	5′-CTGGAACATTAGGCGGAGAA-3′
TmThaumatin-like protein-1_Fw	5′-CTCAAAGGACACGCAGGACT-3′
TmThaumatin-like protein-1_Rv	5′-ACTTTGAGCTTCTCGGGACA-3′
TmThaumatin-like protein-2_Fw	5′-CCGTCTGGCTAGGAGTTCTG-3′
TmThaumatin-like protein-2_Rv	5′-ACTCCTCCAGCTCCGTTACA-3′
TmL27a_qPCR_Fw	5′-TCATCCTGAAGGCAAAGCTCCAGT-3′
TmL27a_qPCR_Rv	5′-AGGTTGGTTAGGCAGGCACCTTTA-3′
TmSpz4-qPCR-Fw	5′-GGCGATGCTCTTCCAGGAC-3′
TmSpz4-qPCR-Rv	5′-CGCGTTCACTCCTTTCATTTGG-3′
TmSpz4-T7-Fw	5′-TAATACGACTCACTATAGGGT CCAGATGTACTGTCGCGATG-3′
TmSpz4-T7-Rv	5′-TAATACGACTCACTATAGGGT TTCCTTCTGTACCAGTCGGG-3′
TmSpz6-qPCR-Fw	5′-GCACAACTCCAAGACGACCT-3′
TmSpz6-qPCR-Rv	5′-TCTCTTCACCCGATCGTTGC-3′
TmSpz6-T7-Fw	5′-TAATACGACTCACTATAGGGT ACCGCGCAAGAGAGTAAAAA-3′
TmSpz6-T7-Rv	5′-TAATACGACTCACTATAGGGT ACGTATCTCCACACCCCTTG-3′
TmSpz3-qPCR-Fw	5′-TCTCAACAACGGGACCTTCG-3′
TmSpz3-qPCR-Rv	5′-GGGACGCCCCGTATGTATTC-3′
TmSPZ3-T7-270bp-Fw	5′-TAATACGACTCACTATAGGGT CGAGAACAAGGCACTGATGA-3′
TmSPZ3-T7-270bp-Rv	5′-TAATACGACTCACTATAGGGT GCGGTGCCATTTGTACTTCT-3′
TmSPZ-like-qPCR-Fw	5′-CAGTTGAGGGTGCCTGTTCA-3′
TmSPZ-like-qPCR-Rv	5′-TTGTTGGCATCGTCCCTTGA-3′
TmSpz-like-T7-Fw	5′-TAATACGACTCACTATAGGGT ATGTTCCCAAAATCAACGGA-3′
TmSpz-like-T7-Rv	5′-TAATACGACTCACTATAGGGT AATCACACGCAGATCCTTCC-3′
dsEGFP_Fw	5′-TAATACGACTCACTATAGGGT CGTAAACGGCCACAAGTTC-3′
dsEGFP_Rv	5′-TAATACGACTCACTATAGGGT TGCTCAGGTAGTGTTGTCG-3′

^1^ Underline indicates T7 promotor sequences.

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
