# Peer review of "Biosurfactants Induce Antimicrobial Peptide Production through the Activation of TmSpatzles in Tenebrio molitor"

_ijms, 2020, doi:10.3390/ijms21176090_

Round 1

Reviewer 1 Report

In this article the authors analyze the immune activation properties of some biosurfactants in the mealworm Tenebrio molitor. By using in vitro assays they demonstrate that the hemolymph of larvae treated with biosurfactants is able to reduce the growth of Gram negative bacteria. Moreover they show that biosurfactants induce AMP expression in the hemocytes, thus increasing the resistance of the larva to bacterial infections. Finally, through gene knockout experiments, they demonstrate that the effects of biosurfactants are mediated by Spatzle and Toll activation.

The present study adds knowledge on the action of biosurfactants in insects. In addition, it is well conceived and the rationale is clear. However, I think that some points should be revised and reconsidered:

  • The authors monitored the expression of 14 genes coding for AMPs. Since these peptides show different behaviors in the various treatments, it is difficult to appreciate a direct correlation between the different Spz isoforms and the various AMPs.   

  • Timing of analysis is not always clear. Along the manuscript the authors generally indicate 24 hours as time point for the analysis, but in the Figures the experiments appear to be monitored up to 10 days.

  • Line 51. Please detail the use of biosurfactants as biopesticides.

  • Figure 1. Statistics should be clearly indicated. What does the line indicate? I suppose that different colors indicate different bacteria, but this must be clearly specified in the Figure legend.

  • Lines 146-149. While the lack of response in the midgut could be due to the administration procedure (injection in the hemolymph), it is difficult to understand the downregulation or lack of variation in AMP expression in the fat body, since this is a key organ responsible for AMP production in insects.

  • Lines 164-168. This paragraph is not clear. Please rephrase.

  • Figure 3. I suggest to indicate the name/acronym of each biosurfactant in the different panels.

  • Paragraph 2.4. How can biosurfactants increase the expression of Spz genes? In addition, why should this happen? Do the authors suggest that biosurfactants both increase the expression of Spz genes and activate the cleavage of Spz? If so, which are the mechanisms involved in both processes?

  • Lines 195-200. While here it seems that these results refer to AMP expression in the whole body, in M&M the authors say that three separate tissues (I suppose hemocytes, fat body, and midgut) were analyzed (see lines 431-432).

  • Lines 330-331. Can the authors at least present some hypotheses about these mechanisms?

  • Lines 373-374. Please provide more information about preliminary experiments that led to choose these concentrations for the current experiments.

  • Paragraph 4.5. The quantity of tissue used for RNA extraction must be indicated.

  • Line 411. Please define “young instar larvae”.

  • Paragraph 4.9 and 4.10 are identical.

  • References should be formatted according to the instructions for authors.

  • The manuscript contains many typos.

Author Response

Rebuttal letter to reviewer’s comments on our submitted manuscript

(ijms-881356)

We received the reviewer comment for the submitted manuscript (ijms-881356). We are sincerely grateful for the reviewer’s positive and constructive comments, which have been very helpful to improve the quality of our paper. The suggestions have noted and we have worked on the reviewer comments. We have taken care of all comments and use track change to revise our manuscript. We hope the manuscript has improved with the suggested revision and would be a novel statement to the scientific community covered by the journal.

Please find the author’s comments to reviewers queries as under:

< Review 1 >

  1. Reviewer comment: The authors monitored the expression of 14 genes coding for AMPs. Since these peptides show different behaviors in the various treatments, it is difficult to appreciate a direct correlation between the different Spz isoforms and the various AMPs.   

Author’s response: we sincerely appreciate the reviewer’s comment. The reviewer comment is very appreciable and it is true that, at this point it is difficult to know the correlation of Spatzle and different AMPs. To clarify this point, we are working on the separately project and we hope that we will come up with clear research result that shows the correlation of Tmspatzle and AMPs induction. 

  1. Reviewer comment: Timing of analysis is not always clear. Along the manuscript the authors generally indicate 24 hours as time point for the analysis, but in the Figures the experiments appear to be monitored up to 10 days.

Author’s response: Thank you for your comment. It is well known that the invaded pathogens activates host immunity and produces effector molecules such as AMPs. In addition, invaded pathogens have to multiply itself to kill the host or to combat with host. Based on these background, we hypothesized that the initial immune activation such as AMPs is important for successful defense against invaded microorganisms. Thus we investigate the induction patterns of TmSpatzle and AMP genes at 24 h post injection of biosurfactants, and monitor the mortality for 10 days post challenge of pathogens. Therefore, the time for different analysis in this study was different with gene as well as purpose of the analysis.

  1. Reviewer comment: Line 51. Please detail the use of biosurfactants as biopesticides.

Author’s response: Thank you for your comment. The use of the biosurfactants as biopesticides have been explained in the next paragraph in introduction part (line number 53 to 60)

  1. Reviewer comment: Figure 1. Statistics should be clearly indicated. What does the line indicate? I suppose that different colors indicate different bacteria, but this must be clearly specified in the Figure legend.

Author’s response: Thank you for your valuable comment. We have clearly stated the statistical significance level and figure legend in the figure (Please refer figure 1).

  1. Reviewer comment: Lines 146-149. While the lack of response in the midgut could be due to the administration procedure (injection in the hemolymph), it is difficult to understand the downregulation or lack of variation in AMP expression in the fat body, since this is a key organ responsible for AMP production in insects.

Author’s response: We would like to thank you for your constructive comment. We strongly agree with the reviewer about the fat body. However, most of the immune related studies use live microorganisms to characterize the immunity, which induce AMPs in different tissues, however, in our case, we injected four different biosurfactants (molecules form) and investigated the induction patterns of AMPs.

The major concern here is how the host recognizes these lipopetides or glycolipids, to activate immune response. Even though the known pattern recognition receptors of D-amino acids have not been identified, D-amino acids are asymmetrically recognized by some other receptors and enzymes in mammals to provide either direct toxic responses or indirect actions through the modulation of antimicrobial peptides (This explanation has been addressed in the discussion part…please refer line number 305-309). Therefore, we carefully suggest that the specific molecules in the used biosurfactants may affect the expression of AMPs in different tissues.

  1. Reviewer comment: Lines 164-168. This paragraph is not clear. Please rephrase.

Author’s response: Thank you for the comment. We have tried to rephrase the paragraph as “However, larvae challenged by E. coli at both 0 and 12 hrs post immune activation by biosurfactant did not show significant survivability, when compared with the control group. This may suggest that AMP expression was regulated 24 hrs post biosurfactant-immune activation. Thus survivability result confirms the AMP expression observed in this study.” (Please refer line number 164-168)

  1. Reviewer comment: Figure 3. I suggest to indicate the name/acronym of each biosurfactant in the different panels.

Author’s response: Thank you for the suggestion. For the clarity and to effectively manage the graph we prefer to use the letter and define as footnote and figure. We hope that this does not affect the quality of the graph.

  1. Reviewer comment: Paragraph 2.4. How can biosurfactants increase the expression of Spz genes? In addition, why should this happen? Do the authors suggest that biosurfactants both increase the expression of Spz genes and activate the cleavage of Spz? If so, which are the mechanisms involved in both processes?

Author’s response: we sincerely thank you for your valuable comment. The mechanism how biosurfactants increase the expression of Spatzle in insect is not well known. However, there is an assumption based on the mammalian model and previous insect immunology studies. We tried to bring here the ideas we used in the discussion part to support our result. For the detailed information please refer discussion part line number 304-330.

“Toll receptors have been identified and known to recognize evolutionarily conserved bacterial components such as LPS, LP, LPT, and PGN in mammals. Moreover, it has been reported that CD14 facilitates LP recognition by toll-like receptors 2 and 1 in humans. Similarly, LPS binding protein (LBP), the serum factor is known to have the ability to bind pathogens and initiates the innate immune respons. Moreover, it has been reported that LP could enhance TLR2-mediated activation, a promising strategy for vaccine development.

Contrarily, in insects, during both embryogenesis and immunity, inactive spätzle protein is synthesized and secreted from the cell, containing a pro domain and a C-terminal region. During infection, microbe associated molecular patterns, such as LPS and lipoteichoic acid (LTA), are recognized by PGN recognition protein and GNBP1, leading to the initiation of the proteolytic cascade, serine protease, and SPE. Therefore, this process finally activates the initially inactive spätzle. Extrapolating to the mammalian and insect immune activation systems, the bacillus LP (fengycin and iturin A) and glycolipid (rhamnolipid) are found activating the TmSpatzle genes in T. molitor. It should be noticed that the recognition receptor for these specific biosurfactants is not well known.”

  1. Reviewer comment Lines 195-200. While here it seems that these results refer to AMP expression in the whole body, in M&M the authors say that three separate tissues (I suppose hemocytes, fat body, and midgut) were analyzed (see lines 431-432).

Author’s response: we thank you for the valuable comments. Under this experiment we did AMP expression analysis from the whole body of spatzle gene silenced larval group. Therefore, we have corrected on the material and method (Please refer line number 446-447)

  1. Reviewer comment: Lines 330-331. Can the authors at least present some hypotheses about these mechanisms?

Author’s response: we thank you for the valuable comments. Concerned the about the hypothesis on receptors of the biosurfactants in the T. molitor, at this point we could not present any hypothesis because it needs extra experiment. To answer this question, we are planning to develop a project to know which receptor is responsible in the recognition of biosurfactant and thus, activate the cleavage of TmSpatzle.

  1. Reviewer comment: Lines 373-374. Please provide more information about preliminary experiments that led to choose these concentrations for the current experiments.

Author’s response: thank you for the comment. We have given the detailed information in the in the material and method (Please refer line number 374-376).Briefly, we conducted the preliminary experiment with 100, 200, 300 and 1000 ng/µl biosurfactants injection and the highest AMPs activity was achieved at 300 ng/µl”

  1. Reviewer comment: Paragraph 4.5. The quantity of tissue used for RNA extraction must be indicated.

Author’s response: Twenty-four hours after biosurfactants injection, the whole body samples from 20 larvae those injected with each biosurfactant in to 500 μl of Guanidine thiocyanate RNA lysis buffer…….. we have included in the material and method part. (Please refer line number 392-393)

  1. Reviewer comment: Line 411. Please define “young instar larvae”.

Author’s response: Thank you for your comment. In mealworm, we normally classify the larval instars in early (1-9 instars), young (10-12), late (13-19) and prepupal larval stage. We have specified in the material method (please refer line number 412)

  1. Reviewer comment: Paragraph 4.9 and 4.10 are identical.

 Author’s response: we sincerely thank you for your constructive comment. We have deleted 4.10.

  1. Reviewer comment: References should be formatted according to the instructions for authors.

 Author’s response: the journal asked to rearrange the reference in number order. So used the number order and formatted the reference. We hope the formatting we used matches the journal requirement.

Reviewer 2 Report

The authors studied the biosurfactant induced antimicrobial activity in the Tenebrio molitor.

Major points:

  1. In chapter 2.1 (pg3) authors show that surfactants have antibacterial activity. Authors claim that this activity is due to AMPs. This activity is not necessary due to AMP presence (clearly shown by authors letter in the paper), because another factors could be involved in antibacterial action: for example, non-AMP insect toxins or applied bio-surfactants, which could be antibacterial by themselves. Thus in chapter 2.1. please change the statements that the effect is due exclusively by AMPs.
  2. Very important point: the PBS control. In this study the PBS injection is considered as a control. But some of injected active surfactants are normally must be dissolved in non-aqueous diluent (for my information, the Sigma-Aldrich Fengycin must be dissolved in MeOH or DMSO; Inturin A in EtOH; Rhamnolipids are required high-alkalinity). Thus, specify in which solvent authors are dissolved the active bio-surfactants (now this very important information is missing in Methods)! In case of EtOH or DMSO usage, the right control is not PBS, but solvent injection.
  3. Figure 5. High level of Ten2 and some other AMPs are induced by control dsEGFP treatment comparing to PBS control, which have the value 1 in this figure. Based on this abnormally high level, the silencing treatment results decreased. Please, explain / speculate about this phenomenon.
  4. For the statistics, line 414, pg 17, “The experiment was performed in triplicates”. Does it mean 3 larvae individuals were involved? Please, specify better the “triplicate”.
  5. Line 426, pg 17: for silencing procedure, just “injecting dsRNA” is reported. Please, specify the manufacturer source of dsRNS and the literature reference for silencing efficiency by this method for this organism.

Minor points:

  1. Line 80, pg 2, typo: Poly-Llysine must be poly-L-lysine? Line 115, pg 3, typo: polypeptides (in plural now) must be polypeptide?
  2. Figure 1. The meaning of each of tree columns is not signed. Which one is gram-positive (S. aureus), gram- negative bacteria (E. coli) or fungi (C. albicans)?
  3. Figure 1. In the text “control group” (line 124, pg 3) is mentioned, but in Figure legend it is not specified. Is it corresponding 100% Bacterial CFU where dotted line is used? Some signs or explanation of this dotted line is need.
  4. Figure 1. In the Legend, “significantly reduced” is written: please report the significance asterisks in the plot.
  5. Line 147, pg 4: “slightly downregulated” is not right, some expressions are strong downregulated, like Ten2 in fat bodies. Thus, correct the statement and please comment/speculate about the downregulations (because the stimulation is expected).
  6. Figure 3 legend, line 172, pg 6: the comma before fengycin and specification of panel C after inturin A are missing.
  7. Figure 4 legend, line 188: commas are missing
  8. Methods: ng/µl or µg/µl ? In line 373, pg15 the bio-surfactant concentration is 300 ng/µl reported, but in line 390 the concentration 300 µg/µl is reported. Is it correct?
  9. The chapters 4.9 and 4.10 in pg 17 are completely equal. It is not correct.

Author Response

Rebuttal letter to reviewer’s comments on our submitted manuscript

(ijms-881356)

We received the reviewer comment for the submitted manuscript (ijms-881356). We are sincerely grateful for the reviewer’s positive and constructive comments, which have been very helpful to improve the quality of our paper. The suggestions have noted and we have worked on the reviewer comments. We have taken care of all comments and use track change to revise our manuscript. We hope the manuscript has improved with the suggested revision and would be a novel statement to the scientific community covered by the journal.

Please find the author’s comments to reviewers queries as under:

< Review 2 >

Major points:

  1. Reviewer comment: In chapter 2.1 (pg3) authors show that surfactants have antibacterial activity. Authors claim that this activity is due to AMPs. This activity is not necessary due to AMP presence (clearly shown by authors letter in the paper), because another factors could be involved in antibacterial action: for example, non-AMP insect toxins or applied bio-surfactants, which could be antibacterial by themselves. Thus in chapter 2.1. please change the statements that the effect is due exclusively by AMPs.

Author’s response: we sincerely thank you for your valuable comments. We accept the personal doubt of the reviewer on the probability of the effect of non- AMP insect toxins. In this line, we did not limit the effect exclusively by AMPs in our discussion part. But, based on our experimental result, we can surely say that the major effect is due to AMPs expression.

  1. Reviewer comment: Very important point: the PBS control. In this study the PBS injection is considered as a control. But some of injected active surfactants are normally must be dissolved in non-aqueous diluent (for my information, the Sigma-Aldrich Fengycin must be dissolved in MeOH or DMSO; Inturin A in EtOH; Rhamnolipids are required high-alkalinity). Thus, specify in which solvent authors are dissolved the active bio-surfactants (now this very important information is missing in Methods)! In case of EtOH or DMSO usage, the right control is not PBS, but solvent injection.

Author’s response: Thank you for your appropriate comment. Concerned this point, since we are working with immune related gene, we worried about the effect of these solvents (MeOH, DMSO and EtOH) on the survivability as well as immune related gene expression, and we did the pre experiment study to check the solvents effect via injecting these solvents at different concentration (25%, 50%, 75% and 100%). Finally, we found that the solvents considerably affect the survivability of the larvae. This effect directly related to immune related genes expression. Therefore, it was not possible to use these solvents as dissolver, because they affect the gene expression as well as survivability.

  1. Reviewer comment: Figure 5. High level of Ten2 and some other AMPs are induced by control dsEGFP treatment comparing to PBS control, which have the value 1 in this figure. Based on this abnormally high level, the silencing treatment results decreased. Please, explain / speculate about this phenomenon.

Author’s response: To clarify this experiment, we inject biosurfactant (or PBS as a control) into TmSpz3 or TmSpz-like-silenced T. molitor larvae. And the values were normalized by PBS group for dsEGFP or dsTmSpzs injected larvae. Thus, the blue bar (dsEGFP) and the orange bar (dsSpzs) indicated the expression level of biosurfactants in control and TmSpz-silenced larvae, respectively. Thus, we hypothesized that the down-regulation in orange bar indicated positive regulation of these AMPs by TmSpzs. Interestingly, several AMPs such as Def2 and Atta1a in figure 5D were up-regulated by dsTmSpz-like treatment. Basically we don’t know why, however, we carefully hypothesized that these expression was affected by larval homeostasis or negative regulation.

  1. Reviewer comment: For the statistics, line 414, pg 17, “The experiment was performed in triplicates”. Does it mean 3 larvae individuals were involved? Please, specify better the “triplicate”.

Author’s response: We appreciate the reviewer’s comment. In the survivability study we used 10 larvae/replication which means 30 larvae in three replication. We have specified in the material and method part. (Please refer line number 412-416)

  1. Reviewer comment: Line 426, pg 17: for silencing procedure, just “injecting dsRNA” is reported. Please, specify the manufacturer source of dsRNS and the literature reference for silencing efficiency by this method for this organism.

Author’s response: Thank you for the comment. We have synthesized the dsRNA of each gene following the protocol. We have included the detailed protocol under paragraph 4.8 (please refer line number 425-438 of the revised version).

Minor points:

  1. Line 80, pg 2, typo: Poly-Llysine must be poly-L-lysine? Line 115, pg 3, typo: polypeptides (in plural now) must be polypeptide?

Author’s response: Thank you for correcting. We have corrected the typo.

  1. Figure 1. The meaning of each of tree columns is not signed. Which one is gram-positive (S. aureus), gram- negative bacteria (E. coli) or fungi (C. albicans)?

Author’s response: We thank you for your comment. We have clarified under figure 1 as foot note and figure.

  1. Figure 1. In the text “control group” (line 124, pg 3) is mentioned, but in Figure legend it is not specified. Is it corresponding 100% Bacterial CFU where dotted line is used? Some signs or explanation of this dotted line is need.

Author’s response: We thank you for your comment. We have specified the control group (Please refer line number 124 and 133-134).

  1. Figure 1. In the Legend, “significantly reduced” is written: please report the significance asterisks in the plot.

Author’s response: we thank you for your comment. We have included the significance astericks in the plot (Please refer figure 1)

  1. Line 147, pg 4: “slightly downregulated” is not right, some expressions are strong downregulated, like Ten2 in fat bodies. Thus, correct the statement and please comment/speculate about the downregulations (because the stimulation is expected).

Author’s response: We thank you for your comment. From our previous experience, the fold down/up regulation is low compared to the PBS injected group, thus we hope that just downregulate is better than strong downregulation. We have deleted the word slightly.

  1. Figure 3 legend, line 172, pg 6: the comma before fengycin and specification of panel C after inturin A are missing.

Author’s response: We thank you for your comment. We have added comma and panel C. (Please refer line number 172)

  1. Figure 4 legend, line 188: commas are missing

Author’s response: We thank you for your comment. We have added comma. (Please refer line number 188)

  1. Methods: ng/µl or µg/µl ? In line 373, pg15 the bio-surfactant concentration is 300 ng/µl reported, but in line 390 the concentration 300 µg/µl is reported. Is it correct?

Author’s response: We thank you for your comment. It is 300 ng/µl not 300 µg/µl. we have corrected to 300 ng/µl (Please refer line number 391)

  1. The chapters 4.9 and 4.10 in pg 17 are completely equal. It is not correct.

Author’s response: We sincerely thank you for your constructive comment. We have deleted 4.10.

Round 2

Reviewer 2 Report

One crucial major point critics is not replayed adequately by authors in the manuscript text: 1 in which solvent (or is it powder?) the biosurfactants are acquared from Sigma Aldrich? 2 in which solvent each of them is dissolved prior the injection to insect? The importance of this information is very high, as authors confirmed the strong immune stimulation by the solvents.

Author Response

Rebuttal letter to reviewer’s comments on our submitted manuscript

(ijms-881356)

We received the 2nd reviewer comments for the submitted manuscript (ijms-881356). We have taken care of all comments and use track change to revise our manuscript. We hope the manuscript has improved with the suggested revision and would be a novel statement to the scientific community covered by the journal.

Reviewer’s comments: 1)One crucial major point critics is not replayed adequately by authors in the manuscript text: 1) in which solvent (or is it powder?) the biosurfactants are acquired from Sigma Aldrich?

Authors response: we sincerely thank you for your valuable comment. We have explained in the material and method as we acquired the biosurfactants from sigma Aldrich in powder form (please refer line number 359 of the revised version).

 Reviewer’s comments: 2) in which solvent each of them is dissolved prior the injection to insect? The importance of this information is very high, as authors confirmed the strong immune stimulation by the solvents

Authors response:  We sincerely thank you for your valuable comment. We normally dissolved by vortexing the biosurfactants in 1x PBS (1x phosphate buffered solution) as solvent. Even though it is not completely dissolved in the water, our target is to deliver the particle as emulsion form in to the larval hemolymph. To insure the uniform dispersion, we vortexed the solution well at each injection. We have explained in the idea in the material and method of the current revised version of the manuscript (please refer line number 373-376). In addition, for the normalization of our experiments, we injected each biosurfactant into T. molitor over 10 larvae as a group.

We hope that with the necessary corrections, our manuscript will be considered suitable for publication in the journal 'IJMS'.

Sincerely,

Dr. Yeon Soo, Han

(Corresponding Author)

Round 3

Reviewer 2 Report

-